# The Characteristics of Milk Fatty Acid Profile Predicted by Fourier-Transform Mid-Infrared Spectroscopy (FT-MIRS) in Chinese Holstein Cows

**DOI:** 10.3390/ani14192785

**Published:** 2024-09-26

**Authors:** Chunfang Li, Haitong Wang, Yikai Fan, Zengpo Zhou, Yuanbao Li, Shengchao Liang, Yabin Ma, Shujun Zhang

**Affiliations:** 1Key Laboratory of Agricultural Animal Genetics, Breeding and Reproduction of Ministry of Education, Huazhong Agricultural University, Wuhan 430070, China; chunfangli0521@webmail.hzau.edu.cn (C.L.); htw0411@webmail.hzau.edu.cn (H.W.); fanyikai123@webmail.hzau.edu.cn (Y.F.); 2Hebei Livestock Breeding Station, Shijiazhuang 050060, China; zhouzengpo@163.com (Z.Z.); 18134178053@163.com (Y.L.); liangshengchao8734@163.com (S.L.); 3Frontiers Science Center for Animal Breeding and Sustainable Production, Huazhong Agricultural University, Wuhan 430070, China; 4The Technology Innovation Center of Cattle Germplasm Resources in Hebei Province, Shijiazhuang 050060, China

**Keywords:** Chinese Holstein cows, FT-MIRS, fatty acid group, fatty acid content, milk characteristics

## Abstract

**Simple Summary:**

Various studies have confirmed that the parity, days in milk (DIM), and somatic cell score (SCS) affected the fatty acid contents in milk and the Fourier-transform mid-infrared spectroscopy (FT-MIRS). However, the characteristics of fatty acid group contents predicted by FT-MIRS in Chinese Holstein cows have not been clearly demonstrated. This study employed FT-MIRS data to study the influencing factors and alterations in milk fatty acid contents in Chinese Holstein cows from large-scale dairy farms in Northern China. This study provided new ideas for the expansion of Dairy Herd Improvement laboratory determination indicators in China and the breeding of dairy cows with excellent milk quality.

**Abstract:**

Fatty acid is an important factor affecting the nutritional quality of milk. In this study, we collected and assessed 78,086 milk samples from 12,065 Chinese Holstein cows from 11 farms in Northern China from November 2019 to September 2022. The contents of eight fatty acid groups were predicted using FT-MIRS-based models. The contents of TFAs, SFAs, UFAs, MUFAs, PUFAs, and LCFAs in milk reached the highest at 96–125 DIM, and SCFA and MCFA contents reached the highest at 276–305 DIM. With the increase in somatic cell score, the contents of various fatty acid groups in milk gradually decreased, and the nutritional value of milk and flavor of dairy products gradually deteriorated. The contents of high-quality fatty acids in milk, particularly UFAs and MUFAs, were significantly higher in the non-pregnant state than in the pregnant state. However, SCFA and MCFA contents exhibited the opposite pattern. Our findings provided valuable information on the content and distribution range of fatty acid groups in milk from Chinese Holstein cows. Further analysis is warranted to explore the breeding of Chinese Holstein cows providing milk with abundant beneficial fatty acids.

## 1. Introduction

The composition and content of nutrients in raw milk are important factors affecting the nutritional value of milk. Among them, fatty acids (FAs) are one of the important nutrients [1]. The FAs and their contents in milk or milk products have attracted much attention, considering their important effects on human health. Some studies have indicated that saturated FAs (SFAs) affected plasma cholesterol, which might be related to atherosclerosis (AS) and coronary heart disease in humans. AS, a chronic inflammatory condition in which atheroma accumulates within the intima of the arterial wall, is responsible for most deaths in Western countries [2]. However, some studies have reported no direct relationship between SFA intake and heart disease [3]. In addition, some studies reported that not all SFAs could increase cholesterol levels, and some exhibited various beneficial biological activities such as anticancer, anti-inflammatory, and antibacterial [4,5,6]. Monounsaturated FAs (MUFAs) can lower blood cholesterol, triglycerides, and low-density lipoprotein cholesterol levels, and their content can also reflect the individual energy and health status of dairy cows [7,8]. Furthermore, the MUFA profile is directly related to the flavor and physicochemical properties of milk and the quality of cheese produced from it [9]. Moreover, polyunsaturated FAs (PUFAs) are closely related to human health. Therefore, as MUFAs and PUFAs are considered beneficial to human health [10], consumers expect milk to have high contents of these unsaturated FAs (UFAs). 

The content and profile of FAs in milk are also related to the health and reproductive status of dairy cows. The C2–C5 in short-chain FAs (SCFAs) in milk are mainly derived from the fermentation by rumen bacteria. Gross et al. [11] reported that ketosis in dairy cows causes a reduction in the contents of SCFAs and medium-chain FAs (MCFAs) in milk. β-hydroxybutyric acid, a derivative of SCFA, is a clinical biomarker for diagnosing ketosis. Researchers are currently exploring FA biomarkers associated with metabolic diseases and reproductive performance in dairy cows [12].

Traditionally, the FA content and composition in milk are usually determined using methods including gas chromatography, ultraviolet spectrophotometry, and liquid chromatography. However, these traditional methods exhibit certain disadvantages, such as high cost, low efficiency, and difficulty in their quick application in batch production. Fourier-transform mid-infrared spectroscopy (FT-MIRS) is a cost-effective detection tool that has been used for the determination of milk protein, fat, and lactose percentages for Dairy Herd Improvement (DHI). Many researchers have employed FT-MIRS to study FAs in milk, e.g., for the analysis of influencing factors, genetic evaluation, heritability, and gene mining [13,14,15]. However, only a few reports are available on Chinese Holstein cows.

The breeds, feeding conditions, and management environment of Chinese Holstein dairy cows are different from those of cows in other countries. Therefore, it is of great significance to analyze the content and proportion of FAs in milk produced by Chinese Holstein dairy cows. Hence, in this study, we aimed to analyze the composition of fatty acid groups in the Northern Chinese Holstein dairy cows and investigate the effects of various factors [including days in milk (DIM), pregnancy state, and somatic cell score (SCS)] on them. Moreover, we explored the alteration pattern of contents of FA groups related to human nutrition and the flavor of dairy products. 

## 2. Materials and Methods

### 2.1. Animal Management and Milk Samples

In this study, lactating Chinese Holstein dairy cows (*n* = 12,065) from 11 large-scale dairy farms in Northern China were selected. All dairy cows were fed a TMR diet, and the cows were milked three times daily at 07:00, 15:00, and 23:00.

Milk samples were collected from all healthy lactating cows at 5–305 DIM monthly during the experimental period (December 2019 to July 2022). During the morning milking, a flow meter (Tru-test, Auckland, New Zealand) was used to collect the evenly mixed milk, and approximately 35 mL of each milk sample was collected. The milk samples were added to new cylindrical bottles (diameter 3.5 cm, height 9 cm). A preservative (0.35 μL bronopol) was immediately added to milk sampling bottles, which were numbered in order. All milk samples were immediately cooled with ice bags after collection and stored at 4 °C until analysis.

### 2.2. Milk Characteristics

Milk samples were sent to the Hebei Livestock Breeding Station within 24 h of collection. All samples were analyzed for fat, protein, lactose, urea, solid not-fat (SNF), total solids (TS), and somatic cell count (SCC) using CombiFOSS FT+ (MilkoScan^TM^7RM, Foss, HillerØd, Denmark), and the corresponding FT-MIRS data were obtained. The SCC was converted into the SCS according to the formula SCS = log_2_(SCC/100) + 3 [16].

### 2.3. Prediction of Fatty Acid (FA) Group Content

In our previous study, using the FA reference values and FT-MIRS data, prediction models for determining the content of eight FA groups were established. The FA groups included total FAs (TFAs), SFAs, UFAs, MUFAs, PUFAs, SCFAs (4–10 carbons), MCFAs (11–16 carbons), and long-chain FAs (LCFAs; 17–24 carbons) (Table 1). The contents of these eight FA groups were determined using the established FA group content prediction models based on the FT-MIRS data. The cross-validation determination coefficients (R_cv_^2^) and root mean square error (RMSE_cv_) for prediction models are presented in Table 2. The FT-MIRS-based prediction models of milk FA content could predict the content of various FA groups in Chinese Holstein dairy cow milk samples with moderate to high accuracy.

### 2.4. Statistical Analysis

SAS 9.0 was used for statistical analysis. The effects of DIM, parity, SCS, pregnancy state, and calving season on FA group content were evaluated and compared using the following mixed linear models: y_ijklmno_ = DIM_i_ + parity_j_ + SCS_k_ + RS_l_ + season_m_ + herd-test date_n_ + cow_o_ + e_ijklmno_
where y_ijklmno_ is the content of TFAs, SFAs, UFAs, MUFAs, PUFAs, SCFAs, MCFAs, and LCFAs and is expressed as g/100 g of fat; DIM_i_ is the fixed effect of the i^th^ DIM (i = 1, 5–35 DIM; i = 2, 36–65 DIM; i = 3, 66–95 DIM; i = 4, 96–125 DIM; i = 5, 126–155 DIM; i = 6, 156–185 DIM; i = 7, 186–215 DIM; i = 8, 216–245 DIM; i = 9, 246–275 DIM; i = 10, 276–305 DIM); parity_j_ is the fixed effect of the j^th^ parity of dairy cows (j = 1, the first parity; j = 2, the second parity; j = 3, the third parity and higher parities); SCS_k_ is the fixed effects of the k^th^ SCS state (k = 1, SCS ≤ −1; k = 2, −1 < SCS ≤ 0; k = 3, 0 < SCS ≤ 1; k = 4, 1 < SCS ≤ 2; k = 5, 2 < SCS ≤ 3; k = 6, 3 < SCS ≤ 4; k = 7, 4 < SCS ≤ 5; k = 8, SCS > 5); RS_l_ is the fixed effect of the l^th^ pregnancy state (l = 1, non-pregnant; l = 2, pregnant); season_m_ is the fixed effect of the m^th^ calving season (m = 1, spring: March–May; m = 2, summer: June–August; m = 3, autumn: September–November; m = 4, winter: December–February); herd-test date_n_ is the random effect of the n^th^ pasture test day (*n* = 1–238); cow_o_ is the random effect of individual dairy cows; e_ijklmnop_ is a random residual assumed to obey e_ijklmno_~N (0, σ_e_^2^) normal distribution, where σ_e_^2^ is the residual variance.

## 3. Results and Discussion

### 3.1. Characteristics of Milk Composition in Chinese Holstein Cows 

Milk composition plays a decisive role in the quality and yield of dairy products [17]. Therefore, dairy companies are very concerned about the relevant indicators in fresh milk, such as fat and protein percentages. In this study, 78,086 milk samples collected from 12,605 dairy cows from 11 farms were analyzed for milk composition, milk yield, and FA properties (Table 3). The average percentages of fat, protein, lactose, SNF, and TS were 3.94%, 3.32%, 5.14%, 9.10%, and 13.07%, respectively (Table 3). The milk fat, lactose, and SNP percentages were slightly higher than those reported in other studies [18,19]. Moreover, the average daily milk yield (35.63 kg/day) was higher than that reported in the literature. The average SCS was 3.42, indicating that some dairy cows were at risk of subclinical mastitis [20]. Therefore, dairy farmers need to improve the management of cows with mastitis. The urea content was 12.39 mg/100 mL, which was slightly lower than that reported by Henao-Velasquez et al. [21] (17.28 mg/dL) and Ma et al. [22] (13.32 mg/dL). The low urea content in this study may be because of certain differences between the TMR diets of dairy cows in Northern China and in developed areas [23].

The results indicated that the TFA content in milk was 40.26 g/100 g of fat. The SFAs and MCFAs accounted for a relatively large proportion (71.48% and 51.56%, respectively) of the TFAs in milk, which was consistent with previous studies [24]. The LCFAs and SCFAs accounted for 36.88% and 8.82% of the TFAs, respectively. Sauer et al. [25] reported that the UFAs in milk were mainly derived from the biohydrogenation of UFAs in the diet, and their absorption was low. Therefore, the proportion of UFAs, MUFAs, and PUFAs in the TFAs was relatively low (27.25%, 23.82%, and 3.63%, respectively), which was consistent with previous studies [19,26]. 

### 3.2. Effects of Fixed Factors on Milk FA Groups

Previous studies reported that FAs are an important source of milk flavor. The even-numbered carbon chain FAs from C4–C12 have a greater impact on milk flavor [27], and these FAs are MCFAs and SCFAs. For example, SCFAs (including butyric acid, caproic acid, caprylic acid, and capric acid) can produce a desirable buttery and creamy taste [28]. DIM, SCS, and pregnancy states extremely significantly affected the contents of SCFAs, MCFAs, and LCFAs (*p* < 0.01; Table 4). 

Modern medical research has reported that the types and contents of FAs in food are closely related to human health [29,30,31]. The uneven proportion of various FAs, particularly the excessive intake of SFAs, can lead to dyslipidemia and further to hypertension, AS, and coronary heart disease. Currently, the effects of SFAs on human health are controversial [32,33]. However, as one of the best strategies to prevent cardiometabolic disease, the U.S. and international dietary guidelines advocate limiting SFA intake [34,35]. Three types of UFA groups (UFAs, MUFAs, and PUFAs) are high-quality FAs in milk. High-quality FAs can decrease blood pressure, increase high-density lipoprotein, and prevent arteriosclerosis. Of them, PUFAs have certain preventive and therapeutic effects on some inflammations and cancers [36,37,38]. Our results indicated that DIM extremely significantly affected the contents of UFAs, MUFAs, and PUFAs (*p* < 0.01). SCS extremely significantly affected MUFA and PUFA contents (*p* < 0.01), which was consistent with the study by Stocco et al. [39]. SCS significantly affected UFA content (*p* < 0.05). Because of the complexity of the biological pathways involved in FA synthesis, we hypothesized that the reason might be that SCC was mainly associated with the desaturation process in the mammary gland [40]. The pregnancy state had no significant effect on PUFA content but had a highly significant effect on UFA and MUFA contents (*p* < 0.01). 

Therefore, DIM significantly affected the content of various FA groups in milk. This might be attributed to the fact that the physiological functions of dairy cows vary with the change in DIM, which has a significant impact on the production performance and content of nutrients in milk [41]. The results indicated that only DIM had an extremely significant (*p* < 0.01) effect on the contents of TFAs and SFAs. The contents of eight FA groups were significantly different at different DIM (*p* < 0.01). The contents of eight FA groups in milk were not significant among different calving seasons and parities (*p* > 0.05), which was inconsistent with previous studies [42,43]. This could be due to the differences in feed delivery method, frequency of feed delivery, type of bedding, frequency of manure removal, and barn design [44]. 

### 3.3. Alteration Tendency of FA Group Contents

#### 3.3.1. The Alteration Tendency of FA Group Contents at Various DIM

The contents of three high-quality FA groups (UFAs, MUFAs, and PUFAs) in milk exhibited a trend of “increase–decrease–increase–decrease” with the extension of DIM, and milk was rich in high-quality FA groups at 96–125 DIM than at 126–155 DIM (*p* < 0.05; Table 5). To our knowledge, at the beginning of lactation, the cow is in a negative energy balance, and its body fat is excessively mobilized with a large amount of LCFAs (C16:0, C18:0, and C18:1 cis-9) in the blood entering the milk [45], whereas the contents of FA groups with different saturation degrees are low in milk. With the progression of lactation, the energy of dairy cows tends to become balanced, and the mobilization of body fat is decreased so that the compositions of FAs in milk become stable. Therefore, in the middle stage of lactation (96–125 DIM) in this study, the produced milk had a higher content of high-quality FA groups. To be able to continuously produce milk rich in high-quality FA groups, TMR diet formulation and feeding management can be combined at 36–155 DIM to effectively produce milk with high nutritional value [46,47,48].

The content of SCFAs exhibited a trend of “increase–decrease–increase” with the extension of DIM. The content of MCFAs exhibited a wave-like change trend of “increase–decrease” throughout the lactation period. The LCFA content reached the highest at 66–95 DIM and then gradually decreased with the extension of DIM. Moreover, this downward trend continued until 276–305 DIM. This indicated that the milk flavor was poor at 126–155 DIM because the contents of SCFAs and MCFAs were the lowest; however, LCFA content was higher. As DIM increased, SCFA and MCFA contents gradually increased and reached the maximum at 276–305 DIM. However, LCFA content reached the minimum at 276–305 DIM, suggesting that the best milk flavor was observed in the late stage of lactation. The LCFA content was maximum at 66–95 DIM, that is, the peak lactation period. This could be because dairy cows needed to obtain energy from body fat in the early lactation period, resulting in decreased FA synthesis by the mammary gland and increased LCFA content in milk [49]. The lactation period, along with energy balance, significantly affects milk FA composition [24,50]. Therefore, the alteration of individual milk FA content in individual cows could be used as the indicator of the energy status of cows [51]. 

#### 3.3.2. The Alteration Tendency of FA Group Content at Different SCS

Our results indicated that SCS had no significant effect on the contents of TFAs and SFAs but had a significant effect on the contents of other FA groups (*p* < 0.05) (Table 6). At SCS ≤ 3, the contents of UFAs and MUFAs in milk exhibited a decreasing trend with the increase in SCS, and both reached the lowest value at 2 < SCS ≤ 3. With the deterioration of mammary gland health of dairy cows, UFA and MUFA contents in milk gradually increased and reached the highest at 4 < SCS ≤ 5. The results indicated that the contents of UFAs and MUFAs were higher at high SCS than at low SCS. At SCS ≤ 5, PUFA content in milk did not change significantly with the increase in SCS; however, at SCS > 5, PUFA content significantly decreased and reached the lowest. Pecka-Kiełb et al. [52] reported a negative correlation between SCS and fat content. SCS exhibited no significant effect on SFA and UFA, which was also observed in the present study. However, they did not report the corresponding SCS level; therefore, it is difficult to compare for which further study is needed to determine the SCS threshold. Ma et al. [53] reported that milk with a high SCC has more fat hydrolysis, and the milk is prone to FA oxidation; this leads to diminished milk flavor and nutritional value, ultimately reducing the economic value of milk. Therefore, we suggest that Chinese dairy companies should include low SCS in the pricing standard and increase the purchase price of low-SCS milk to increase the enthusiasm of dairy farmers. 

At SCS ≤ 4, the contents of SCFAs and MCFAs in milk increased and further decreased with the increasing SCS. SCFA and MCFA contents were the lowest at SCS < 0 and reached the highest at 2 < SCS ≤ 3. At SCS > 4, SCFA and MCFA contents tended to decrease with the increase in SCS. Additionally, LCFA content gradually decreased as SCS increased and reached the highest at SCS ≤ −1 and the lowest at SCS > 5. These results indicated that the contents of SCFAs, MCFAs, and LCFAs decreased at SCS > 4 compared with those at 3 < SCS ≤ 4. This could be attributed to the deterioration of cow health because of mastitis, which decreased the dry matter intake of the cows and ultimately led to the decrease in the contents of SCFAs, MCFAs, and LCFAs. Thus, it is essential to recognize subclinical diseases in cows in time to intervene and prevent the further development of diseases. It can be determined based on the characteristics of the changes in the contents of SCFAs, MCFAs, and LCFAs in milk under the conditions of various influencing factors [12]. 

Therefore, at SCS ≤ 5, high-SCS milk contained a higher content of FA groups, particularly UFAs, MUFAs, SCFAs, and LCFAs, than low-SCS milk. At SCS > 5, the contents of FA groups in milk decreased; particularly, PUFA and LCFA contents exhibited the lowest values. This is consistent with the reports by Stocco et al. [39] and Pegolo et al. [40]. 

#### 3.3.3. The Alteration Tendency of FA Group Content under Different Pregnancy States

Pregnancy states significantly affected the contents of most FA groups (except TFAs, SFAs, and PUFAs) (*p* < 0.05) (Figure 1). The content of SFAs in milk in the non-pregnant state was lower than that in the pregnant state, but the contents of UFAs, MUFAs, and PUFAs were higher in the non-pregnant state than in the pregnant state. The contents of UFAs and MUFAs were significantly higher in the non-pregnant state than in the pregnant state (*p* < 0.05). Therefore, cows in the non-pregnant state produced milk with higher nutritional value. 

The contents of SCFAs and MCFAs were significantly higher in the pregnant state than in the non-pregnant state (*p* < 0.05). However, LCFA content was lower in the pregnant state, which resulted in better milk flavor during the pregnant state. Therefore, it is necessary to strengthen the feeding and management levels of cows in the pregnant state to improve the welfare of cows and obtain more milk with high-quality dairy flavor. 

These results indicated that the contents of the FA groups varied with the pregnancy states of dairy cows, which was consistent with the studies by Bastin et al. [54] and Toledo-Alvarado et al. [14]. Stádník et al. [55] and Ntallaris et al. [56] reported that the changes in the composition and percentage of FAs in milk could be used to predict the energy states and reproductive ability of Holstein dairy cows. This could be because the change in reproductive status would promote the body to regulate energy allocation and endocrine function, thus leading to changes in metabolites in blood or milk. Therefore, further research is needed to investigate these effects.

## 4. Conclusions

DIM, SCS, and pregnancy state significantly affected the content of FA groups (*p* < 0.05). From the perspective of human nutrition and health, the quality of milk produced in the middle stage of lactation was better than that in the late stage of lactation. The quality of milk from cows in the non-pregnant state was significantly better than that from cows in the pregnant state. Based on milk quality and economic benefits, it is recommended that large-scale farms in Northern China focus on high milk yield, high milk fat and protein percentages, and high-quality nutrient levels such as low SFA and high UFA contents in milk. Our findings expanded DHI measurement indicators with FAs for Chinese dairy cows and laid a foundation for the breeding of dairy cows producing milk rich in high-quality FAs, as well as the efficient production of specialty milk sources. 

## Figures and Tables

**Figure 1 animals-14-02785-f001:**
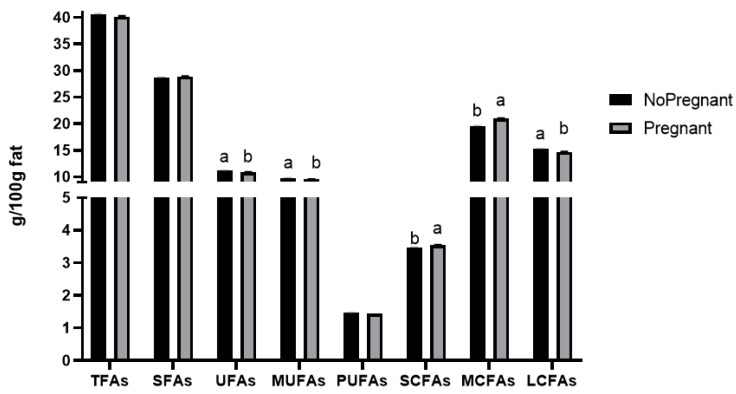
Effect of pregnancy states on the content of FA groups in milk. Different superscript letters (a, b) indicate significant differences at *p* < 0.05.

**Table 1 animals-14-02785-t001:** Fatty acid (FA) groups and their composition.

Trait	Included Fatty Acids
TFAs	SFAs, UFAs, SCFAs, MCFAs, LCFAs
SFAs	C4:0, C6:0, C8:0, C10:0, C11:0, C12:0, C13:0, C14:0, C15:0, C16:0, C17:0, C18:0, C20:0, C21:0, C22:0, C23:0, C24:0
UFAs	MUFAs, PUFAs
MUFAs	C14:1 cis-9, C15:1 cis-10, C16:1 cis-9, C17:1 cis-10, C18:1 cis-9, C18:1 trans-11, C20:1 cis-11, C22:1 cis-13, C24:1 cis-15
PUFAs	C18:2 cis-9,12, C18:2 trans-9,12, C18:3 cis-6,9,12, C18:3 cis-9,12,15, C20:2 cis-11,14, C20:3 cis-8,11,14, C20:3 cis-11,14,17, C20:4 cis-5,8,11,14, C20:5 cis-5,8,11,14,17, C22:2 cis-13,16, C22:6 cis-4,7,10,13,16,19
SCFAs	C4–C10
MCFAs	C11–C16
LCFAs	C17–C24

**Table 2 animals-14-02785-t002:** Cross-validation determination coefficients (R_cv_^2^) and root mean square error (RMSE_cv_) of prediction models of FA groups.

Model	R_cv_^2^	RMSE_cv_
TFAs	0.6991	9.2437
SFAs	0.7096	6.5024
UFAs	0.6505	3.1345
MUFAs	0.6475	2.7831
PUFAs	0.6798	0.4021
SCFAs	0.7239	0.9693
MCFAs	0.6643	5.1758
LCFAs	0.6192	4.1199

**Table 3 animals-14-02785-t003:** Descriptive statistics for FAs, milk composition, and milk yield.

Fatty Acids	Mean ± SD	% of TFAs	Milk Composition	Mean ± SD
TFAs (g/100 g of fat)	40.26 ± 13.75		Fat (%)	3.94 ± 0.82
SFAs (g/100 g of fat)	28.78 ± 9.77	71.48	Protein (%)	3.32 ± 0.38
UFAs (g/100 g of fat)	10.97 ± 4.37	27.25	Lactose (%)	5.14 ± 0.23
MUFAs (g/100 g of fat)	9.59 ± 3.94	23.82	SNF (%)	9.10 ± 0.43
PUFAs (g/100 g of fat)	1.46 ± 0.55	3.63	TS (%)	13.07 ± 1.07
SCFAs (g/100 g of fat)	3.55 ± 1.47	8.82	Urea (mg/100 mL)	12.93 ± 3.17
MCFAs (g/100 g of fat)	20.76 ± 7.42	51.56	Milk yield (kg/day)	35.63 ± 10.77
LCFAs (g/100 g of fat)	14.85 ± 5.78	36.88	SCS (units)	3.42 ± 3.81

**Table 4 animals-14-02785-t004:** Influence of parity, DIM, pregnancy state, calving seasons, and SCS on the contents of FA groups.

Fatty Acid Groups	Parities		DIM		Pregnancy States		Calving Seasons		SCS	
	F-Value	Pr > F	F-Value	Pr > F	F-Value	Pr > F	F-Value	Pr > F	F-Value	Pr > F
TFAs	0.09	0.91	12.3	<0.01	2.98	0.08	1.68	0.17	1.38	0.21
SFAs	0.13	0.88	19.10	<0.01	1.06	0.30	1.58	0.19	0.92	0.49
UFAs	0.29	0.75	9.56	<0.01	17.24	<0.01	1.46	0.22	2.3	0.02
MUFAs	0.36	0.70	9.36	<0.01	18.79	<0.01	1.46	0.22	3.46	0.00
PUFAs	0.54	0.58	18.43	<0.01	2.38	0.12	1.40	0.24	2.82	0.01
SCFAs	1.57	0.21	11.41	<0.01	8.45	0.00	1.26	0.29	4.34	<0.01
MCFAs	0.77	0.47	41.46	<0.01	107.62	<0.01	1.11	0.34	11.75	<0.01
LCFAs	1.40	0.25	24.08	<0.01	32.53	<0.01	1.61	0.19	12.71	<0.01

Note: *p* < 0.05, significant; *p* < 0.01, extremely significant.

**Table 5 animals-14-02785-t005:** Effects of different DIM stages on the contents of FA groups in milk.

DIM	TFAs	SFAs	UFAs	MUFAs	PUFAs	SCFAs	MCFAs	LCFAs
5–35	39.33 ± 0.22 c	28.26 ± 0.16 b	10.53 ± 0.07 c	9.17 ± 0.06 c	1.44 ± 0.01 bc	3.43 ± 0.02 b	20.32 ± 0.12 bc	14.9 ± 0.09 b
36–65	39.99 ± 0.21 b	28.52 ± 0.15 b	10.84 ± 0.07 b	9.47 ± 0.06 b	1.45 ± 0.01 bc	3.48 ± 0.02 b	20.22 ± 0.12 bc	15.2 ± 0.09 ab
66–95	40.28 ± 0.18 b	28.62 ± 0.13 b	10.99 ± 0.06 b	9.6 ± 0.05 b	1.46 ± 0.01 bc	3.44 ± 0.02 b	19.77 ± 0.09 c	15.38 ± 0.07 a
96–125	41.36 ± 0.21 a	29.42 ± 0.15 a	11.36 ± 0.07 a	9.92 ± 0.06 a	1.52 ± 0.01 a	3.57 ± 0.02 ab	20.36 ± 0.11 bc	15.29 ± 0.09 ab
126–155	39.53 ± 0.21 c	27.76 ± 0.15 b	10.98 ± 0.07 b	9.62 ± 0.06 b	1.42 ± 0.01 c	3.41 ± 0.02 b	19.08 ± 0.11 d	15.19 ± 0.09 ab
156–185	40.62 ± 0.21 b	28.93 ± 0.15 b	11.24 ± 0.07 a	9.85 ± 0.06 a	1.45 ± 0.01 bc	3.52 ± 0.02 ab	20.57 ± 0.11 b	14.83 ± 0.09 b
186–215	40.62 ± 0.21 b	28.85 ± 0.15 b	11.17 ± 0.07 ab	9.77 ± 0.06 ab	1.48 ± 0.01 b	3.52 ± 0.02 ab	20.04 ± 0.11 c	15.03 ± 0.09 b
216–245	41.09 ± 0.21 a	29.25 ± 0.15 a	11.24 ± 0.07 a	9.84 ± 0.06 a	1.48 ± 0.01 b	3.53 ± 0.02 ab	20.54 ± 0.11 b	14.96 ± 0.09 b
246–275	40.93 ± 0.21 ab	29.17 ± 0.15 ab	11.24 ± 0.07 a	9.84 ± 0.06 a	1.47 ± 0.01 bc	3.55 ± 0.02 ab	20.61 ± 0.12 b	14.83 ± 0.09 b
276–305	40.08 ± 0.21 b	28.94 ± 0.15 b	10.85 ± 0.07 b	9.47 ± 0.06 b	1.44 ± 0.01 c	3.59 ± 0.02 a	21.17 ± 0.11 a	14.1 ± 0.09 c

Note: Results are given as least square mean ± standard errors. Different lowercase letters (a, b, and c) in the same column indicate significant differences (*p* < 0.05).

**Table 6 animals-14-02785-t006:** Effects of various DIM stages on the contents of FA groups in milk.

SCS	TFAs	SFAs	UFAs	MUFAs	PUFAs	SCFAs	MCFAs	LCFAs
≤−1	40.63 ± 0.6	28.99 ± 0.43	11.04 ± 0.19 ab	9.66 ± 0.17 ab	1.47 ± 0.02 a	3.4 ± 0.06 b	19.47 ± 0.32 c	15.36 ± 0.25 a
−1 < SCS ≤ 0	40.27 ± 0.24	28.7 ± 0.17	11.04 ± 0.08 ab	9.64 ± 0.07 ab	1.47 ± 0.01 ab	3.42 ± 0.03 b	19.66 ± 0.13 c	15.01 ± 0.10 a
>0, ≤1	40.32 ± 0.13	28.78 ± 0.09	11.03 ± 0.04 ab	9.63 ± 0.04 ab	1.47 ± 0.01 a	3.53 ± 0.01 a	20.38 ± 0.07 b	15.14 ± 0.05 a
>1, ≤2	40.28 ± 0.12	28.73 ± 0.08	11.00 ± 0.04 ab	9.60 ± 0.03 b	1.46 ± 0.00 a	3.55 ± 0.01 a	20.56 ± 0.06 ab	15.13 ± 0.05 a
>2, ≤3	40.23 ± 0.12	28.72 ± 0.09	10.95 ± 0.04 b	9.56 ± 0.03 b	1.46 ± 0.00 ab	3.55 ± 0.01 a	20.65 ± 0.06 a	15.02 ± 0.05 a
>3, ≤4	40.32 ± 0.13	28.69 ± 0.09	11.07 ± 0.04 ab	9.68 ± 0.04 ab	1.46 ± 0.01 ab	3.55 ± 0.01 a	20.65 ± 0.07 ab	14.86 ± 0.05 a
>4, ≤5	40.75 ± 0.15	28.96 ± 0.11	11.17 ± 0.05 a	9.78 ± 0.04 a	1.47 ± 0.01 ab	3.54 ± 0.02 a	20.58 ± 0.08 ab	14.76 ± 0.06 ab
>5	40.28 ± 0.18	28.62 ± 0.13	11.05 ± 0.06 ab	9.71 ± 0.05 ab	1.44 ± 0.01 b	3.51 ± 0.02 a	20.19 ± 0.10 b	14.48 ± 0.08 b

Note: Results are given as least square mean ± standard errors. Different lowercase letters (a, b, and c) in the same column indicate significant differences (*p* < 0.05).

## Data Availability

The datasets used and/or analyzed during the current study are available from the corresponding author upon reasonable request.

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
