# Peer review of "The Characteristics of Milk Fatty Acid Profile Predicted by Fourier-Transform Mid-Infrared Spectroscopy (FT-MIRS) in Chinese Holstein Cows"

_animals, 2024, doi:10.3390/ani14192785_

Round 1
Reviewer 1 Report
Comments and Suggestions for Authors
Dear Editor,
Here is my review on the manuscript with the number 3131822-Animals
Thanks for the cooperation.
Regards
Similarity index: 24%

Dear Editor,
Here is my review on the manuscript with the number 3131822-Animals
Thanks for the cooperation.
Regards
Similarity index: 24%
Author Response
Dear editor and reviewer:
Thank you for reviewing our manuscript and providing valuable feedback. We greatly appreciate your contribution and support to our research. Following your suggestions, we have carefully revised and adjusted the manuscript to enhance its quality and readability. Below are our responses to each of the points you raised:
Comments 1:The criticism of this study is that there are many research articles on this subject.The subject is the same: Fourier Transform Mid-Infrared Spectroscopy and working with different animal species. The nutritional contents of the milk of different animals and milk urea nitrogen are being investigated. In this respect, this study is not an original study. It is not an innovative research..
Response 1: Thanks to the reviewer for the question. As you said, although there have been relevant studies, this study is not innovative. But the unique breeding mode of Chinese Holstein dairy cows (large scale, herd farming, TMR feeding) makes the study of great significance to study the nutritional composition of Chinese Holstein dairy cows. Meanwhile, there are few relevant studies on the fatty acid content of Chinese Holstein dairy cows' milk by using FT-MIRS. Therefore, this study can provide a reference for the research on the characteristics of fatty acids in Chinese Holstein cows,which is of great significance.
Comments 2: The entire text must be checked for English from the very beginning.
Response 2: Thank you for pointing this out. We agree with the reviewer's comments and have checked for English from the very beginning. For example P1 lines 26-30 of the manuscript.
Comments 3: It has been observed that although a space should be left after the last sentences in the text, this was not done. This rule should be corrected throughout the text.
Response 3: Thank you for pointing this out. We agree with this comment. Therefore, we have checked all the last sentences in the text to see if there are any spaces left, and modified them according to the rules. For example P2 lines 46-56 of the manuscript.
Comments 4: The reference in the following heading of the material method section is incorrect:
2.3. Prediction of fatty acid group content.
Response 4: Thank you for pointing this out. We agree to amend and correct the corresponding parts of the whole text. For example P4 line 151 of the manuscript.
Comments 5: Which package program was used in the statistical analysis section should be written (SPSS, SAS, etc???).
Response 5: Thank you for pointing this out. The statistical methods in this study have been described, see line 120 of the manuscript.
Comments 6: The study looked at the urea in milk (mean 12.39 mg/100 ml). What is the nitrogen value of milk urea? Because milk urea nitrogen is the biggest indicator of whether feeding is done correctly in a study.
Response 6: Thanks to the reviewer for the question. In this study, the average urea content in cow milk was 12.39 mg/100 ml, and the nitrogen value of milk urea was 26.55 mg/100 ml. In addition, the average urea content in cow milk is within the recommended standard range in the United States (10-14 mg/100 ml) and Canada (8-14 mg/100 ml). Although there is no standard for the range of urea content in milk in China, the range recommended by farmers in cow production is 10-18 mg/100 ml, which indicates that the cattle feeding in this study is correct.
Comments 7: Is the milk fat level (3.94%) you obtained in the trial high?
Response 7: Thanks to the reviewer for the question. The average milk fat percentage of cattle in this study was 3.94%, which was not high. According to China's "National Standard for Food Safety Raw Milk" (GB19301-2010) and "Technical Specification for Raw Milk Use Classification" (TTD-STIA001-2019), the milk fat percentage of regular milk is 3.1-4%.
Comments 8: The fat-free solids of milk vary within certain limits. This value is important in determining adulteration of milk. This issue should be given more importance and the value obtained in the study should be discussed with the values found in other research.
Response 8: Thank you for your suggestions.We have added the discussion, as detailed in lines 146-147of the manuscript.
Comments 9: Why was the milk density value not examined in the study? Density is a criterion that gives information about whether milk is adulterated or not.
Response 9: Thanks to the reviewer for the question. This study used the milk composition detector was CombiFOSS FT+ (MilkoScanTM7RM, Foss, HillerØd, Denmark),which had not purchase milk density value detection model. The milk composition detector only had modles of fat, protein, lactose, urea, solid not-fat (SnF), total solids (TS), and somatic cell count (SCC).
Comments 10: 3.1. Characteristics of Milk Composition in Chinese Holstein Cows: There is not enough discussion. 3.3.2. The Change Tendency of Fatty Acid Groups Content among Different SCS: There is not enough discussion.
Response 10: Thank you for pointing these out. We have enriched the discussion, see lines 140-154, lines 242-250, and lines 258-260 of the manuscript, respectively.
Comments 11: There is a mistake Page 7.
Response 11: Thank you for pointing these out. We have fixed the mistake on Page 7 and updated the reference, see the line 251 of the manuscript.
Comments 12: The Characteristics of the keyword is changed to milk Characteristics.
Response 12:Thank you for your suggestion. We have fixed, see in line 37 of the manuscript.
Comments 13: References are not uniform.
Response 13: Thank you for pointing these out. We have fixed all mistakes of references, see in lines 342-469 of the manuscript.
Comments 14: There are alot of old references..
Response 14: Thank you for pointing these out. According to the content of the article, we deleted the old references and added new ones, also see in lines 342-469 of the manuscript.
Reviewer 2 Report
Comments and Suggestions for Authors
Overall manuscript must be revised professionally for english grammar. It reads confusinf at some points.
Lines 87-89: descibe how milk samples were collected please (waikato? other device?)
Lines 113-132: Fatty acids can be analyzed as an absolut number (gr/d) o relative to total FA (% of total FA). Due to lactation curve of milk production and of milk fat content, it is expected that fatty acids vary greatly, therefore it would be more insightful to analyze and discuss changes in milk fatty profile (as % of total) instead of absolute values. Please include rationale of using absolute values, or altenatively presenta data and discussion based on fatty acid profile as % of total fatty acids.
Lines 155-159: the idea of LDL link to SARA and mamary gland uptake of LCFA is not clear. Please delete or elaborate to get a coherent idea.
161-167: Please clarify that these effects are on humen health.
Comments on the Quality of English LanguagePlease have a professional editing for english languaje included. Many errors and worng wording make the reading not very smooth.
Author Response
Dear editor and reviewer:
Thank you for reviewing our manuscript and providing valuable feedback. We greatly appreciate your contribution and support to our research. Following your suggestions, we have carefully revised and adjusted the manuscript to enhance its quality and readability. Below are our responses to each of the points you raised:
Comments 1: Overall manuscript must be revised professionally for english grammar. It reads confusinf at some points.
Response 1: Thank you for pointing this out. We agree with the reviewer's comments and have checked for English from the very beginning.
Comments 2: Lines 87-89: descibe how milk samples were collected please (waikato? other device?)
Response 2: Thanks to the reviewer for the question. The collection of milk samples is described in lines 89-95 of the manuscript.
Comments 3: Lines 113-132: Fatty acids can be analyzed as an absolut number (gr/d) relative to total FA (% of total FA). Due to lactation curve of milk production and of milk fat content, it is expected that fatty acids vary greatly, therefore it would be more insightful to analyze and discuss changes in milk fatty profile (as % of total) instead of absolute values. Please include rationale of using absolute values, or altenatively presenta data and discussion based on fatty acid profile as % of total fatty acids.
Response 3: Thank you for your suggestions. After discussion, our team decided to use the absolute values to represent the fatty acids content of milk. When our team established the fatty acids models, the overall coefficient of variation in g/100g of fat was large. The average coefficient of variation in g/100g of fat was 57.305%, while the average coefficient of variation in the percentage of total fatty acids was reduced to 26.017%. Considering that relatively larger variation will improve the generalization ability of the prediction model, and g/100g of fat as a unit can calculate the absolute values of fatty acids in milk to determine the accurate values of fatty acids in milk, which is more valuable for the evaluation of production performance and milk quality of cows. Therefore the unit of absolute values is selected for the establishment of the prediction models and subsequent prediction. In additon, many other studies also used absolute values, such as the studies of Haiyue et al., Martini et al. and Zebari et al.
Haiyue H ,Yan T ,Junjie Z , et al.Authentication of organically produced cow milk by fatty acid profile combined with chemometrics: A case study in China[J].Journal of Food Composition and Analysis,2023,120.
Martini M, Altomonte I, Sodi I, Vasylieva Y, Salari F. Sterol, tocopherol, and bioactive fatty acid differences between conventional, high-quality, and organic cow milk. J Dairy Sci. 2023 Dec;106(12):8239-8248. doi: 10.3168/jds.2023-23378.
Zebari HM, Rutter SM, Bleach ECL. Fatty acid profile of milk for determining reproductive status in lactating Holstein Friesian cows. Anim Reprod Sci. 2019 Mar;202:26-34.
Comments 4: Lines 155-159: the idea of LDL link to SARA and mamary gland uptake of LCFA is not clear. Please delete or elaborate to get a coherent idea.
Response 4: Thank you for your suggestions. We have removed this section.
Comments 5: 161-167: Please clarify that these effects are on humen health.
Response 5: Thank you for your suggestions. We explained the relationship between fatty acids and human health, especially the effects of fatty acids on human hypertension and atherosclerosis,see in lines 172-183 of the manuscript.
Comments 6: Please have a professional editing for english languaje included. Many errors and worng wording make the reading not very smooth.
Response 6: Thank you for pointing this out. We agree with the reviewer's comments and have edited for English.
Round 2
Reviewer 1 Report
Comments and Suggestions for Authors
Regards
Author Response
We are very grateful for your comments.
Reviewer 2 Report
Comments and Suggestions for Authors
Why was milk fatty acid profile changed to milk fatty acids profile?
Line 41: change is for are
Line 42: Can you star a sentence with an abbreviation?
Line 42: Besause?
Line 44: change affects for affect
Line 53: change is for are. English note: when using a plural such as "acids", they "are" not, they "is".
Line 59: Should read "SCFA and medium-chain fatty acid (MCFA) content in milk."
Line 63: should read :"FA content and composition"
Line 76: Should read: Hence, our objectIVE was to survEY fatty acid groups composition
Lines 78-79: Should read: "We were also interested in exploring the changing pattern of fatty acid groups content related to human nutrition and dairy product flavor"
Line 86: change "at" for "between"
Lines 89-92 should read: " The milk samples WERE added into cylindrical bottles (diameter 3.5 cm, height 9 cm). Preservative WAS IMMEDIATELY ADDED TO milk sampling bottles (0.35μL of Bronopol) and were numbered in order. All milk samples were immediately COOLED WITH ICE PACKS after collection AND STORED at 4 °C until analysisl.
ENGLISH grammar, typos, and writing ERRORS ARE POINTED OUT UP TO LINE 92. PLEASE HAVE THE REMAINING MANUSCRIPT CAREFULLY READ AND CORRECTED
Define Atherosclerosis (AS) in line 169
Lines 167-177: I somewhat disagree, as research has also shown positive or no effects of SFA coming from dairy in human health. I would expect a critical review/discussion of this.
Line 177: no need to put (AS) as it is defined above.
Comments on the Quality of English LanguageAbove
Author Response
Dear editor and reviewer:
Thank you for reviewing our manuscript and providing valuable feedback. We greatly appreciate your contribution and support to our research. Following your suggestions, we have carefully revised and adjusted the manuscript to enhance its quality and readability. All changes made to the text are in green in the revised manuscript so that they may be easily identified. Below are our responses to each of the points you raised:
Comments 1:Why was milk fatty acid profile changed to milk fatty acids profile?
Response 1: Thanks to the reviewer for the question. Fatty acid is the same as fatty acids, but for the sake of consistency, we changed them to fatty acids.
Comments 2: Line 41: change is for are.
Response 2: Thanks. We have been changed, see line 42 of the manuscript.
Comments 3: Line 42: Can you star a sentence with an abbreviation?
Response 3: Thanks. We agree with this comment and have been changed, see line 43 of the manuscript. Whereafter, we have checked and modified all the similar errors.
Comments 4: Line 42: Besause?.
Response 4: Thanks. We fixed the spelling error, see line 44 of the manuscript.
Comments 5: Line 44: change affects for affect
Response 5: Thanks. We have been done, see line 45 of the manuscript.
Comments 6: Line 53:change is for are. English note: when using a plural such as "acids", they "are" not, they "is".
Response 6: Thank you for your guidance. We have been done, see line 55 of the manuscript. Whereafter, we have checked and modifiedall the similar errors.
Comments 7: Line 59: Should read "SCFA and medium-chain fatty acid (MCFA) content in milk."
Response 7: Thank you for your guidance. In order to more smoother, we have changed this sentence to read "causes reduction in the contents of SCFAs and medium-chain FAs (MCFAs) in milk.", see line 61 of the manuscript.
Comments 8: Line 63: should read :"FA content and composition"
Response 8: Thank you for pointing this out. We have been changed, see line 65 of the manuscript. Thank you again for your guidance.
Comments 9: Line 76: Should read: Hence, our objective was to survey fatty acid groups composition
Response 9: Thank you for pointing these out. In order to keep clarity and flow, we have changed this sentence to read "Hence, in this study, we aimed to analyze the composition of fatty acid groups in the northern Chinese Holstein dairy cows and investigate the effects of various factors [including days in milk (DIM), pregnancy states, and somatic cell score (SCS)] on them. ", see line 78 of the manuscript. Thank you again for your guidance.
Comments 10:Lines 78-79: Should read: "We were also interested in exploring the changing pattern of fatty acid groups content related to human nutrition and dairy product flavor".
Response 10: Thank you for pointing these out. In order to keep clarity and flow, we have changed this sentence to read "Moreover, we explored the alteration pattern of contents of FA groups related to human nutrition and flavor of dairy products ", see lines 81-82 of the manuscript.
Comments 11: Line 86: change "at" for "between".
Response 11: Thanks. We have been done, see line 87 of the manuscript.
Comments 12: Lines 89-92 should read: " The milk samples WERE added into cylindrical bottles (diameter 3.5 cm, height 9 cm). Preservative WAS IMMEDIATELY ADDED TO milk sampling bottles (0.35μL of Bronopol) and were numbered in order. All milk samples were immediately COOLED WITH ICE PACKS after collection AND STORED at 4 °C until analysisl. .
Response 12:Thank you for pointing these out. In order to keep clarity, flow, and easy understanding, we have changed this sentence to read "The milk samples were added to new cylindrical bottles (diameter 3.5 cm, height 9 cm). A preservative (0.35 μL bronopol) was immediately added to milk sampling bottles, which were numbered in order. All milk samples were immediately cooled with ice bags after collection and stored at 4 °C until analysis. " , see lines 91-95 of the manuscript. Thank you again for your guidance.
Comments 13: ENGLISH grammar, typos, and writing ERRORS ARE POINTED OUT UP TO LINE 92. PLEASE HAVE THE REMAINING MANUSCRIPT CAREFULLY READ AND CORRECTED.
Response 13: Thanks. We agree with the reviewer's comments and had it professionally touched up.
Comments 14: Define Atherosclerosis (AS) in line 169.
Response 14: Thank you for your suggestions. We have added the define of Atherosclerosis (AS), as detailed in lines 178- 181 of the manuscript.
Comments 15: Lines 167-177: I somewhat disagree, as research has also shown positive or no effects of SFA coming from dairy in human health. I would expect a critical review/discussion of this.
Response 15: Thanks. We have added the discussion of SFA effects on human health, as detailed in lines 171-176 of the manuscript.
Comments 16: Line 177: no need to put (AS) as it is defined above.
Response 16: Thank you for pointing this out. We have been done, see line 184 of the manuscript. Thank you again for your guidance.
Round 3
Reviewer 2 Report
Comments and Suggestions for Authors
Line 23: should read milk quality instead of quality milk.
Line 44: needs proofreading
Line 85: use at instead of between
Comments on the Quality of English LanguageNeeds a professional proofreading.
Author Response
Dear editor and reviewer:
Thank you for reviewing our manuscript and providing valuable feedback. We greatly appreciate your contribution and support to our research. Following your suggestions, we have carefully revised and adjusted the manuscript to enhance its quality and readability. All changes made to the text are in purple in the revised manuscript so that they may be easily identified. Below are our responses to each of the points you raised:
Comments 1:Line 23: should read milk quality instead of quality milk.
Response 1: Thank you for pointing this out. "quality milk" has been modified to "milk quality", see line 23 of the manuscript.
Comments 2: Line 44: needs proofreading
Response 2: Thanks. We have changed "because" to "considering", so that the sentence is more fluent and understandable.You can see line 44 of the manuscript.
Comments 3: Line 85: use at instead of between
Response 3: Thanks. We agree with this comment and have changed "between" to "at", see line 89 of the manuscript.
Comments 4: the results must be improved.
Response 4: Thank you for your suggestions. After discussion, we deeply understood your suggestions and improved the results and discussion part again. First of all, atherosclerosis (AS) was simplified in the conclusion and discussion part, the definition of AS was put into the Introduction part, and the report about AS was added. Then, the description and discussion of the effects of SCS on milk quality or fatty acids were added to the results and discussion part. The added contents made the full text smoother, making the results also smoother and clearer.
The details are as follows:
- Reintroduced atherosclerosis (AS)in the Introduction of this manuscript, and deleted the content about AS in the conclusion and discussion part, as detailed in lines 46-48 and line 173 of the manuscript.
- Briefly summariz the impact of SFAs on human healthin the conclusion and discussion part, see lines 173-174 of the manuscript.
- Our study found that SCS significantly affected MUFAsand PUFAs, and the results of other studies were consistent with ours (see the line 182 of the manuscript).
- The reasons for the significant effects of SCS on UFAcontent were speculated in the results and disscusion part, as shown in lines 183-185 of the manuscript.
- The reason for the maximum LCFA content at 66-95 DIM wasspeculated in the results and disscusion part, see lines 226-228 of the manuscript.
- Energy imbalance in lactation canaffect FA content in milk, and the change of single fatty acid content can be used as an indicator of energy state of dairy cows. You can see the lines 229-231 of the manuscript.
- Our study found that SCS affected most of FA content, but had no effect on SFA. Other researchers also reported similar results, but they did not report the corresponding SCS level, so it is difficult to compare our study results, seelines 244-248 for details.
Comments 5: Needs a professional proofreading
Response 5: Thank you for your suggestions. According to your advice, this manuscript was edited for proper English language, grammar, punctuation, spelling, and overall style by one or more of the highly qualified native English speaking editors at MJEditor (https://www.mjeditor.com/). MJEditor specializes in editing and proofreading scientific manuscripts for submission to peer-reviewed journals.